# Role of Epithelial to Mesenchymal Transition in Colorectal Cancer

**DOI:** 10.3390/ijms241914815

**Published:** 2023-10-01

**Authors:** Jian Lu, Marko Kornmann, Benno Traub

**Affiliations:** Department of General and Visceral Surgery, Ulm University Hospital, Albert-Einstein-Allee 23, 89081 Ulm, Germany; charleslujian@163.com (J.L.); marko.kornmann@uniklinik-ulm.de (M.K.)

**Keywords:** CRC, EMT, MET, EMP, CSC, CMS, EMT-TF, TME, TGF-β, CTC, tumor heterogeneity, tumor microenvironment, biomarkers, clinical prognosis

## Abstract

The epithelial–mesenchymal transition (EMT) is a cellular reprogramming process that occurs during embryonic development and adult tissue homeostasis. This process involves epithelial cells acquiring a mesenchymal phenotype. Through EMT, cancer cells acquire properties associated with a more aggressive phenotype. EMT and its opposite, mesenchymal–epithelial transition (MET), have been described in more tumors over the past ten years, including colorectal cancer (CRC). When EMT is activated, the expression of the epithelial marker E-cadherin is decreased and the expression of the mesenchymal marker vimentin is raised. As a result, cells temporarily take on a mesenchymal phenotype, becoming motile and promoting the spread of tumor cells. Epithelial–mesenchymal plasticity (EMP) has become a hot issue in CRC because strong inducers of EMT (such as transforming growth factor β, TGF-β) can initiate EMT and regulate metastasis, microenvironment, and immune system resistance in CRC. In this review, we take into account the significance of EMT-MET in CRC and the impact of the epithelial cells’ plasticity on the prognosis of CRC. The analysis of connection between EMT and colorectal cancer stem cells (CCSCs) will help to further clarify the current meager understandings of EMT. Recent advances affecting important EMT transcription factors and EMT and CCSCs are highlighted. We come to the conclusion that the regulatory network for EMT in CRC is complicated, with a great deal of crosstalk and alternate paths. More thorough research is required to more effectively connect the clinical management of CRC with biomarkers and targeted treatments associated with EMT.

## 1. Introduction

Cancer is a significant global public health concern, and colorectal cancer (CRC) is the second-leading cause of cancer-related death [1,2]. It is expected that in 2023, there will be over 150,000 new instances of CRC in the United States [3,4,5]. In recent years, the rate of CRC has risen in China. CRC can be viewed as an indicator of socioeconomic development, and in countries undergoing significant transitions, the incidence rate tends to increase in tandem with the human development index [2,6].

More than half of cases and fatalities can be linked to preventable risk factors such as smoking, a Western diet, excessive alcohol use, the use of antibiotics, inactivity, and obesity. This risk seems to be decreased by calcium supplements, appropriate whole grain, dietary fiber, and dairy product consumption [2,6,7,8,9,10,11]. Consistent increases in the number of cases of early-onset CRC have been observed in developed nations, indicating that similar risk factors and environments within these regions are significant contributors [5,10,12,13].

Strong evidence exists that colon cancer screening, which removes colonic polyps as precursor lesions, lowers colon cancer morbidity and death. Despite this, CRC screening is still not widely used [14,15,16,17,18,19,20]. The primary prevention of CRC continues to be an important method for lowering its rising global burden [2].

A variety of genetic and epigenetic changes gradually compound to form colorectal adenomas and invasive adenocarcinomas in healthy colonic epithelial cells, which is how CRC develops [21]. The pathophysiological effects of epigenetic modifications, particularly DNA methylation and histone modifications, are significant in the onset and advancement of CRC [22,23].

Genetic changes that interfere with the DNA repair machinery are responsible for the pathogenesis of CRC, which results in aberrant crypt formation in the colon. Adenomatous or serrated polyps can evolve into CRC as a result of the accumulation of additional signaling system alterations [24]. An aberrant crypt takes approximately 10–15 years to progress into a polyp, a neoplastic precursor lesion, and ultimately, CRC. Currently, stem or stem-like cells are thought to be the cells of origin for the majority of colorectal malignancies [9].

Heterogeneity of the tumor is a major determinant of tumor biology, treatment response, and patient survival [25,26,27]. The cumulative accumulation of well-known genetic and epigenetic modifications leads to the development of CRC heterogeneity, which is a significant factor in therapy effectiveness. The pathophysiology of CRC has been found to involve three different pathways: chromosomal instability, microsatellite instability (MSI-H), and CpG island methylation phenotype (CIMP) [4,25,27,28,29,30,31,32].

The complicated and tightly regulated process of epithelial–mesenchymal transition (EMT) may be governed by hundreds of transcription factors and thousands of microRNAs [33]. It is believed that metastatic spread and the development of secondary tumors depend on cancer cell EMT plasticity programs overlaid on intrinsic genetic flaws [34]. The progression and development of CRC are driven by the accumulation of somatic mutations in oncogenes and tumor suppressor genes within colonic stem cells [35].

The most comprehensive CRC categorization method to date has been provided by CRC Subtyping Consortium, which claims the identification of four consensus molecular subtypes (CMSs) for a better prognostic evaluation and enhanced development of innovative therapeutics. Multiple cohorts exploring various treatment strategies demonstrate predictive value for CMS categorization in early and late settings [29,36]. Hypermutation, MSI, and a strong immune response are features of CMS1. CMS2 demonstrates epithelial characteristics and a strong stimulation of Wnt and Myc signaling. Epithelial characteristics and metabolic abnormalities are displayed by CMS3. A “mesenchymal” phenotype with a strong transforming growth factor β (TGF-β) activation, improved angiogenesis, stromal activation, and inflammatory infiltration define CMS4 [4,32,37,38,39]. Particularly, the CMS4 categorization was linked to an EMT signature that included a notable decrease in the microRNAs that control tumor suppression via ZEB1 and ZEB2 [40]. This feature may contribute to the explanation of why CMS4 had the worst prognosis out of the four subtypes, along with matrix remodeling and traits associated with TGF-β.

The investigation of gene expression-based subtypes has the potential to create a new paradigm for stratified CRC treatment [27]. The heterogeneity of metastasis, however, still poses a significant issue because the existing framework is predicated on the transcriptome profile of the initial tumor.

## 2. Search Strategy and Selection Criteria

We identified publications from 2008 to 2023 by searching PubMed. Key words searched included “epithelial-mesenchymal transition”, “EMT”, “MET”, “pEMT”, “epithelial plasticity”, “epithelial-mesenchymal plasticity”, “colorectal cancer”, “colon cancer”, “rectal cancer”, “cancer”, “chemotherapy”, “chemoresistance”, “prognosis”, “consensus molecular subtype”, and “CMS”. We concentrated on publications with high-impact factors that accept peer review. Only English-language articles were reviewed.

## 3. Characteristics of EMT

EMT is a biological reprogramming process that allows epithelial cells to take on a mesenchymal phenotype during embryonic development and adult tissue homeostasis. EMT is crucial for the development of the embryo, the healing of wounds, and the advancement of cancer, as it gives cancer cells aggressive phenotype-associated characteristics. Tight junctions dissolve, apical–basal polarity is disrupted, and the cytoskeletal structure is reorganized in tumor cells after EMT is activated. These physical changes promote cell migration away from the original site, invasion of neighboring tissues, metastasis, survival in the bloodstream, and eventually cause metastases to grow in distant organs [40,41,42,43,44,45,46,47,48,49,50,51].

The EMT program does not operate in a strictly binary manner, and cancer cells seldom complete the entire EMT cycle, which would cause them to become totally mesenchymal. EMT is essential for the development of the embryo, the healing of wounds, and the progression of cancer. This is because EMT confers aggressive phenotype-associated traits on cancer cells, which are necessary for the progression of cancer. The latter of these types of junctions are created by the synthesis of epithelial cell adhesion molecules (E-cadherin). E-cadherin maintains the connection between epithelial cells in healthy tissue by acting as a glue. After EMT activation, the expression of mesenchymal markers such as N-cadherin, vimentin, fibroblast-specific protein 1, and fibronectin increased and the expression of epithelial markers such as E-cadherin, ZO-1, and occludin decreased. This causes the epithelial cells to lose their characteristic polygonal, cobblestone morphology (Table 1). Cells undergo a transformation in which they assume a spindle-shaped mesenchymal appearance and have the ability to move [22,44,48,52,53,54].

It is not shocking that EMT has been linked to the spread of tumor cells because it increases cell mobility [46]. EMT works as a dynamic program that alternates between alternate epithelial and mesenchymal phases to produce a variety of cells with different phenotypes. In addition, the fact that the execution of the EMT program can be influenced by the environmental signals that are experienced by cells as well as the pre-existing differentiation lineages of these cells means that EMT cannot be viewed as a singular and stereotypical program. Instead, EMT can be described as a collection of programs that share many elements in common.

Individual cells with a mixture of epithelial and mesenchymal characteristics do indeed seem to be the rule rather than the exception. In both healthy and malignant human tissues, completion of the EMT program and formation of full mesenchymal cells may be uncommon [51]. The vast majority of cancers are epithelial carcinomas, and epithelial tumor cells acquire dedifferentiation, migration, and invasive behavior during the progression from benign adenomas to malignancies and metastases [55]. Emerging evidence suggests that partial EMT (pEMT) can induce distinct migratory properties, thereby enhancing cancer cell epithelial–mesenchymal plasticity and cell fate plasticity [46]. The greatest potential for metastatic spread exists in cancer populations with incomplete EMT. Among these heterogeneous subpopulations, those with a higher proportion of epithelial characteristics and less mesenchymal transition were more malignant and more likely to spread [40,49].

EMT is a reversible biological process that temporarily transforms epithelial cells into a paramesenchymal state. mesenchymal–epithelial transition (MET), a reversal process, allows the extra mesenchymal cells to shift back to the epithelial state [44]. Numerous investigations have pointed to the necessity of reversing EMT as a prerequisite for successful metastatic colonization [22,43,44,48,57,58,59,60]. The environment has a significant impact on the occurrence of EMT and/or MET, with cell type and tissue type playing the most significant roles [60]. In the intricate progression from primary tumor to metastasis, cancer cells must adapt to shifting and frequently unfavorable environmental conditions. This tumor cell plasticity is demonstrated by the reversible transition from differentiated to undifferentiated or partially EMT-associated cancer cell phenotypes [61]. Human CRC cell clusters in an E/M state contribute to metastatic seeding (Figure 1), and the loss of E-cadherin expression during culture frequently induces EMT in pure mesenchymal tumor cells. Thus, the characteristics of local invasion and tumor initiation are attained [62].

Extensive phenotypic heterogeneity can be produced within tumors by individual cells that develop into various states along the E-M lineage, and this phenotypic plasticity and heterogeneity can make cancer cells more adaptive and resistant. The activation of EMT has been observed to enhance the aggressiveness of this particular phenotype and reduce the susceptibility of tumor cells to chemotherapy. This effect is achieved through the modification of the microenvironment, leading to the development of quasi-mesenchymal cells that display heightened resistance to therapeutic approaches like chemotherapy and immunotherapy [22]. EMT-inducible transcription factors (EMT-TFs) ZEB1, SNAIL, and SLUG confer resistance to oxaliplatin- and cisplatin-based chemotherapy in colon cancer [44].

## 4. EMT in Polyps, IBD, and CRC

Premalignant polyp-like lesions are the origin of CRC, and genetic and epigenetic changes gradually accumulate over time to cause the multistep evolution that is seen at the histopathological level. The majority of CRC comes from two precancerous polyp subtypes: the traditional adenoma–carcinoma pathway (also known as chromosomal instability sequence), which accounts for 60–70% of CRC, and the serrated tumor pathway (10–20%) [3,9,18,28,32,63,64,65]. Adenomatous colorectal polyps are thought to take about 17 years to fully develop into primary CRC [47].

Inflammatory bowel disease (IBD) is a chronic, recurrent inflammatory disorder of the gastrointestinal tract that increases the risk of CRC. When CRC was diagnosed in patients with IBD, they were younger, and the tumors developed in the colon’s inflammatory location with typical clinicopathologic characteristics [3,66,67,68,69,70,71,72]. Patients with persistent colonic inflammation run the risk of developing colon cancer, which is a well-known and concerning consequence. Observational studies have repeatedly demonstrated that the risk of developing colitis-associated colorectal cancer (CACRC) is substantially correlated with the degree, duration, and severity of inflammation. Inflammation plays a significant cumulative role in raising IBD-associated CRC risk [73]. Important risk factors for CRC include persistent and poorly controlled IBD and chronic inflammation of the gastrointestinal tract due to poor dietary practices, such as a Western diet [30].

The majority of CRC is believed to originate from precancerous lesions termed adenomas via the adenoma–cancer pathway. Asymptomatic individuals who undergo screening and resection of colonic adenomas can reduce CRC morbidity and mortality [74]. EMT happens in mucosal and skin barriers during inflammation and healing, and persistent EMT has been seen in cancer, asthma, and chronic rhinosinusitis (CRS) [56]. Although the molecular mechanisms of tissue damage-induced EMT are yet unknown, they most likely result from two separate signaling types. Firstly, a wide variety of soluble EMT-inducing side signals, such as tumor necrosis factor (TNF), interleukin-6 (IL-6), and IL-1β, can be produced by damage coupled with a substantial inflammatory response. Alternately, a cell-intrinsic mechanism causing EMT may be activated by the local breakdown of the epithelial cell sheet’s physical continuity and the resulting loss of homotypic cell–cell adhesive contacts [48]. Claudin-7 (Cldn-7) is a member of the tight junction (TJs) family and is strongly associated with cancer and inflammation. It is dedicated to preserving cell polarity and the integrity of the TJs barrier. By compromising the integrity of TJs and escalating the inflammatory cascade, Cldn-7 deficiency encourages colitis and the ensuing malignant transformation [75].

Although patients with IBD can develop sporadic malignancies, chronic inflammation plays a significant role in risk stratification as the underlying cause. CACRC demonstrates a transition from CMS2 to CMS4 (EMT), dysregulation of Wnt signaling favoring TGF-β activation, and a CD4+ cell-rich “immune inflammatory” immunosuppressive microenvironment [69]. Activation of WNT/β-catenin signaling by loss-of-function mutations in the APC gene is a defining characteristic of sporadic CRC. β-catenin-dependent and -independent mechanisms regulate cell proliferation, polarity, and stemness via this pathway. WNT signaling is a key regulator of EMT in CRC and may therefore contribute to the observed mesenchymal bias in inflammatory bowel disease-related colorectal cancer (IBD-CRC) [66].

Partial EMT is present in more than 90% of human CRC cell lines, which is a condition that encourages the formation of cell clusters during CRC spread. EMT is therefore a prospective target for preventing original cancers from becoming aggressive or for preventing metastasis and recurrence following tumor resection. The adaptability and variety of the different pathways involved make it difficult to develop medications that specifically target EMT. To combat drug resistance in metastatic CRC, however, promising therapeutic approaches include: (1) combining EMT inhibitors with conventional chemotherapy medicines; and (2) employing EMT inhibitors in adjuvant therapy to decrease tumor resection relapse [45].

The liver is the most prevalent metastatic site for CRC because it receives and filters blood from the gastrointestinal tract via the portal vein [76]. The effectiveness of immunotherapy in treating metastatic CRC is limited to tumors exhibiting defective mismatch repair and high microsatellite instability, suggesting that tumor cells possess the ability to influence their immune milieu [77]. The formation of metastases is the culmination of a multistep process involving the dissemination of cancer cells to various organs, followed by their adaptation and proliferation in the microenvironment of the foreign tissue. It has been thought that EMT is the first step in the metastatic cascade, particularly in the process of CRC cells acquiring a migratory phenotype [78].

By downregulating E-cadherin and other components of epithelial junctions, CRC cells isolate themselves from one another and from nearby normal or cancer cells in the initial step of the invasion–metastasis cascade. They subsequently penetrate through the underlying stroma and destroy the basement membrane by secreting or locally activating matrix metalloproteinases (MMPs) and urokinase plasminogen activator (u-PA). Proangiogenic factors, including VEGF and other proangiogenic cytokines, are released as a result of the pericellular matrix and extracellular matrix (ECM) degrading. These factors encourage the growth of new lymphatic and blood capillaries, which aids in the infiltration of tumor cells into the bloodstream. Because of this, tumor cells can either enter blood arteries directly or indirectly through colon-lymphatic channels [45].

CRC is a TGF-β-rich cancer, and CRC tumors and cell lines express more TGF-β than normal colonic epithelium [79]. TGF-β is an extensively researched cytokine associated with EMT. More TGF-β signaling has been seen in CAFs with poor prognostic CRC subtypes as part of a stromal profile linked to disease recurrence. TGF-β is the most extensively studied cytokine released by CAFs. Within the GTPase and PI3K, MAPK/ERK, WNT, and AKT/mTOR pathways, TGF-β signaling can initiate EMT by activating EMT-TFs over time [50,54,80].

It is believed that Wnt/β-catenin signaling activation is essential for the induction of EMT in CRC. There is significant impact for reducing CRC growth when the Wnt/β-catenin axis is targeted [81].

## 5. Mechanism of EMT in CRC

Transcriptional regulation, post-translational control, epigenetic modification, and noncoding RNA-mediated regulation are some of the complex network components that closely regulate EMT. Snail, Twist, Zeb, and other EMT-related transcription factors are expressed at the post-transcriptional and post-translational levels under the regulation of many signaling pathways (Figure 2). These pathways also regulate the downstream transcriptional network, which further controls the biological impacts of EMT, in conjunction with other epigenetic variables. They suppress the production of genes that uphold the epithelial state and promote the mesenchymal state [22,40,43,44,46,48,50,51,61,82,83].

Multiple signaling pathways, including TGF-β/SMAD, WNT/β-catenin, Notch, and receptor tyrosine kinase, can be activated by extracellular stimuli from the tumor microenvironment. These signaling pathways induce a core set of EMT transcription factors to initiate the EMT program in tumor cells [88]. The EMT program directed by EMT-TFs can bestow various qualities essential to the malignant evolution of tumor cells, including tumor-initiating traits, motility, dissemination capability, and greater resistance to frequently employed chemotherapeutic medicines [44,89]. Even though a sizeable majority of the research conducted on EMT focuses on transcription factor networks, EMT can also be controlled by post-translational modifications, microRNAs, long noncoding RNAs, post-transcriptional alterations, epigenetic regulators, and alternative splicing. In order to produce these highly flexible cellular states, various levels of regulation do not serve as independent mediators; rather, they constitute a highly interconnected network.

It has been demonstrated that SNAIL plays a crucial role in CRC, and elevated classical WNT signaling in CRC cells raises SNAIL levels, while SNAIL inhibits the E-cadherin-encoding gene CDH1 by binding to the E-box in the CDH1 promoter and recruiting the polycomb repressive complex, thereby regulating EMT and promoting local invasion [44,50,90,91].

SNAIL is also a significant factor in the relationship between EMT and CRC stem cell characteristics since it directly binds to the E3/E4 E-box of IL-8 to activate IL-8 production, which in turn causes cancer stem cell activity and tumorigenicity in human CRC cells [90,92]. EMT-TFs frequently work together to control the expression of shared target genes and frequently also exert control over one another’s expression. For instance, SNAIL is an upstream regulator that increases the expression of numerous additional EMT-TFs, such as SLUG, ZEB1, and TWIST1 [50].

Epithelial gene expression can be downregulated and mesenchymal gene expression can be upregulated by three TFs from the SNAIL family (SNAIL1, SNAIL2, and SNAIL3) and two TFs from the basic helix–loop–helix (BHLH) family (TWIST1 and TWIST2) [22]. EMT is also influenced by the WNT/β-catenin signaling pathway, which is promoted by SNAIL’s interaction with -βcatenin. As a result, it is thought that upregulating SNAIL is a crucial stage in EMT. By triggering increased levels of SNAIL expression, several signaling pathways—NOTCH, TGF-β, and WNT/β-catenin signaling pathways—play a synergistic role in the onset and development of EMT [90].

The stroma of human colorectal malignancies contains cancer cells with a mesenchymal phenotype that are TWIST1-positive. CRC patients had higher TWIST1 mRNA levels than healthy individuals [53]. To facilitate the morphological and behavioral changes necessary for migration, EMT-undergoing cells must activate mesenchymal genes. Strong promoters of the mesenchymal transcriptional program are the EMT-TFs Twist (TWIST1) and Pair-related homeobox 1 (PRRX1) [46].

Similar to SNIAL, TWIST1 can stimulate the production of N-cadherin while suppressing the expression of E-cadherin, thereby increasing cell mobility and decreasing cell adhesion. Several EMT program signaling pathways have an effect on TWIST. It is significant to note that in hypoxic conditions, the transcriptional machinery of the hypoxia-inducible factor-1α (HIF1A) promotes EMT and the spread of tumor cells by activating TWIST expression [22].

Zinc finger E-box binding homeobox 1 (ZEB1), a transcriptional repressor, has recently been demonstrated to increase tumor cell invasion and metastasis. ZEB1 is a key inducer of EMT in a variety of human malignancies. ZEB1 directly inhibits the transcription of the microRNA-200 family members miR-141 and miR-200c, which are known to aggressively promote the epithelial differentiation of CRC cells [22,93,94]. ZEB1 functions as a transcription factor that promotes mesenchymal differentiation and is directly regulated by over 60 miRNA families [33].

By attracting additional chromatin modifiers to the promoters of CDH1 and N-cadherin, ZEB1 similarly to SNAIL transcriptionally suppresses CDH1 and promotes the genes that encode these proteins [44]. By blocking the glycosylase N-methyl-purine glycosylase (MPG), which promotes colitis and inflammation-related CRC in epithelial cells, ZEB1 also offers new therapeutic approaches for controlling inflammation and inflammation-related malignancies [70].

The transcription factor ZEB2, which induces the epithelial-to-mesenchymal transition, is expressed at the invasion front, linked to the advancement of tumors, and serves as a prognostic indicator for CRC. ZEB2 and SNAIL promote the synthesis of MMPs, aid in cell invasion, and encourage the breakdown of basement membranes [44]. ZEB2 expression is associated with a poor oncological prognosis and distant recurrence; hence, incorporating ZEB2 expression status into the tumor–node–metastasis (TNM) staging system facilitates the process of identifying patients who are at high risk of recurrence [95].

Some noncoding RNAs have the ability to control and be controlled by important EMT genes, which can affect the EMT program. The most investigated families of noncoding RNAs are the miR-34 and microRNA-200 (miR-200) families. For instance, miR-200c targets ZEB1 mRNA translation in order to induce epithelial differentiation, thereby inhibiting the migration and invasion of CRC [40,50,94,96]. Through a double-negative feedback loop, miRNAs of the miR-200 family effectively upregulate cellular E-cadherin levels, thereby maintaining a more epithelial phenotype [96,97]. A mesenchymal-like spindle morphology is brought on by miR-200 family inhibition, and this is followed by enhanced cell invasion and migration, which is thought to be a crucial first stage in the metastatic spread of cancer [97].

Due to the importance of ZEB1 and ZEB2 in EMT, the activities of these proteins need to be strictly controlled. The miR-200 family is believed to function as a key regulator of epithelial phenotypes by specifically targeting ZEB1 and ZEB2, which are transcriptional repressors of genes involved in cell adhesion (E-cadherin) and polarity (CRB3 and LGL2). ZEB1/ZEB2 are upregulated and their target genes for cell adhesion and polarity are downregulated when the miR-200 family is epigenetically silenced [98].

Tumor-associated macrophages (TAMs) are a significant component of the tumor microenvironment and are frequently linked to tumor metastasis in human malignancies. TAMs activate an EMT program by controlling the JAK2/STAT3/miR-506-3p/FoxQ1 axis to promote CRC invasion, migration, and CTC-mediated metastasis. Consequently, the activation of this process induces the synthesis of CCL2, thereby facilitating the attraction of macrophages. This discovery unveils a novel form of communication between immune cells and tumor cells inside the microenvironment of CRC [99]. The immunological microenvironment and clinical outcome are significantly influenced by chemokines connected to CRC. The chemokine CXCL12 is crucial for the spread of CRC, whereas CCL2 is intimately linked to the buildup of macrophages in the hypoxic microenvironment of CRC [63].

The canonical Wnt signaling pathway, commonly referred to as the Wnt/β-catenin signaling system, is widely recognized as a prominent signaling route in CRC, playing a crucial role in driving the progression of colon cancer. The sustained activation of Wnt signaling has been observed to contribute to the development of CRC. Additionally, the Wnt/β-catenin signaling pathway has been found to play a crucial role in maintaining the stemness of both normal and malignant cells [37]. The alteration and degradation of β-catenin, a functional Wnt signaling effector molecule, are crucial processes in the Wnt signaling pathway as well as in the onset and development of colon cancer. By sequestering β-catenin into the cytoplasm, this traditional route limits nuclear accumulation of the protein. Upregulation of SNAIL, a key EMT regulator that suppresses E-cadherin and encourages migration and local invasion, is a result of enhanced Wnt signaling. A positive feedback loop governs the crosstalk between Wnt signaling and SNAIL, whereby SNAIL overexpression amplifies the expression of Wnt target genes. Moreover, increased Wnt signaling inhibits GSK3β, preventing GSK3β phosphorylation and β-TrCP-mediated ubiquitination from destroying Slug. Lastly, SNAIL and Slug accumulation also suppresses E-cadherin, a crucial EMT characteristic [40,100,101,102].

In CRC, Notch signaling activation is linked to a short survival, the CSC phenotype, and EMT, which promotes tumor growth [102]. The complex interaction between Wnt and Notch signaling pathways determines the control of intestinal cell fate determination and lineage specification in intestinal stem cells (ISCs) and colonic CSCs. Both pathways support the control of EMT, which is necessary to produce tumor cells from more differentiated tumor cells that resemble stem cells. Either increasing cadherin-dependent β-catenin-mediated transcription via Wnt signaling, or altering TGF-β activity or inducing the NF-κB pathway via Notch signaling, initiates the EMT process [96].

Encoded SMAD4 protein functions as a tumor suppressor and is an important modulator of the TGF-β pathway, which governs cell division. Mutations in SMAD4 can cause uncontrollably multiplying cells, and this mutation is primarily responsible for the EMT and metastatic processes. While Wnt and fibroblast growth factors regulate SMAD4 activity in response to heightened TGF-β signaling, constitutive activation of the TGF-β pathway is not required. In total, 10% to 20% of patients with CRC carry this mutation. SMAD4 mutations additionally predict resistance to oxaliplatin-based chemotherapy and contribute to a poor prognosis [71].

Elevated expression of miR-4775 stimulated EMT in vitro and in vivo by activating TGF-β signaling through Smad7. It also facilitated the invasion and metastasis of CRC cells. This was reversed by overexpressing Smad7 or downregulating miR-4775 [103].

Additional studies revealed the role and mechanism of RHOJ (a small GTPase that is preferentially expressed in EMT cancer cells) as a key regulator of EMT-associated chemoresistance [104]. CircRNAs are expressed abnormally in CRC and are associated with the clinicopathological characteristics and prognosis of CRC patients. Certain circRNAs control EMT by promoting circPTK2 or inhibiting circSMAD7 [105].

## 6. The Clinical Role of EMT

In terms of tumor response, progression-free survival (PFS), and overall survival (OS), anti-EGFR monoclonal antibody with chemotherapy is the most effective treatment for patients with metastatic colorectal cancer (mCRC) whose primary tumor is situated in the left colon or rectum. Regarding this, unique gene changes and expression profiles found in mCRC originating in the right colon determine a decreased sensitivity to EGFR inhibition. Features of EMT, including overexpression of TGF-α, enhanced receptor tyrosine kinase (AXL) and tyrosine protein kinase receptor (EPHA2) signaling, and increased TGF-β signaling, have been associated with resistance to anti-EGFR drugs [4].

EMT phenotypic heterogeneity in human malignancies is still not standardized for assessment, despite its relevance in drug resistance and metastasis; EMT-IFA (immunofluorescence assay) can be used clinically to monitor tumor adaptation to therapy [106]. The timely collaboration of a complex network of regulators and molecular signaling pathways is necessary for the EMT process. Three categories of factors can be identified from these factors: transcription factors that coordinate the EMT program (called EMT core regulators), effector molecules that carry out the EMT program (called EMT effectors), and extracellular signals that initiate the EMT program (called EMT inducers) [40,90]. These linkages between EMT and CRC reveal a complicated network of relationships between them, which presents significant difficulties for therapeutic therapy.

EMT has a significant impact on tumor development, metastasis, and medication resistance in CRC. There is growing proof that EMT indicators can act as outcome predictors and possible treatment targets in CRC from preclinical and early clinical research [40]. E-cadherin downregulation, which causes adhesion junctions to become unstable, is a crucial component in EMT. In stage III CRC, a poor prognosis is linked to loss of E-cadherin expression. It has been discovered that CRC patients have an elevated risk of cancer recurrence and a decreased chance of survival due to aberrant regulation of transcription factors connected to EMT and mesenchymal markers. According to earlier research, the aggressiveness, metastasis, and poor prognosis of CRC are all connected to the overexpression of transcription factors relevant to EMT, such as SNAIL, SLUG, TWIST, and ZEB [90]. Vimentin processes mechanical feedback and controls the dynamics of microtubule and actin networks to aid in the promotion of cell migration. Therefore, there is little question that vimentin’s enhanced expression encourages CRC to invade [40].

EMT caused by elevated mesenchymal gene expression in epithelial tumors is a sign of a poor prognosis in CRC. The expression patterns of both mesenchymal cells present in the tumor microenvironment (TME) and epithelial cancer cells are reflected in the transcriptome of tumor tissues. In the transcriptome-based CRC data, TME cells, particularly cancer-associated fibroblasts (CAFs), were primarily responsible for the expression of mesenchymal genes as opposed to cancer cells [107].

One study has suggested establishing a prognostic model based on the characteristics of EMT-related genes (ERGs) [108]. There is still debate surrounding the conventional prognostic risk assessment of CRC patients in stages II/III. The development of malignant tumors is thought to be intimately related to EMT. Therefore, creating a prognostic model based on EMT seems promising.

Experiments in mice models of skin or breast cancer have shown that the activation of the EMT program in primary tumors is essential for the spread of tumor cells to the lungs. Once disseminated, the cells must undergo MET in order to develop large-scale metastases [50]. It is still unclear how the EMT program’s termination mechanistically affects the metastatic colonization procedure. Additionally, not all metastases demand the entire restoration of epithelial characteristics. Similarly, in the process of CRC liver metastasis, cancer cells metastasize from the primary tumor, invade the liver locally, and then colonize the liver. The EMT/MET mechanism implicated in this situation must be investigated further in the future.

The presence of several protein markers linked to EMT can serve as a highly accurate predictor of high-grade cancer [44]. With the help of single-cell RNA sequencing and cell surface markers, various EMT transition phases can be recognized. Different EMT transition stages perform unique roles, with mixed EMT states having the greatest potential for metastatic spread. Different EMT transition phases exhibit various chromosomal and gene expression patterns [109].

The majority of invasive epithelial carcinoma cells go through EMT. Cancer cells in the mesenchymal and intermediate phases group together with other cells in the microenvironment to form clumps, invade blood arteries, and develop into CTCs. CTCs facilitate the spread of CRC to the liver, lungs, and lymph nodes. CTCs isolated from orthotopic CRC xenograft models generated organoids (CTCs-derived organoids, CTCDOs). CTCDO exhibited a mixed EMT state and increased expression of stemness-related markers. Functionally, CTCDOs showed higher migration/invasion abilities and different responses to pathway-targeting drugs compared with xenograft-derived organoids (XDOs) [49,110]. It has also been reported that adjuvant therapy for colon cancer patients guided by circulating tumor DNA (ctDNA) is beneficial [111].

Noninvasive liquid biopsy is presently the most effective technical tool for improving clinical decision-making in oncology. It can be analyzed multiple times and monitored in real time for tumor recurrence, metastasis, and treatment response. The identification of CTC, ctDNA, exosomes, and tumor education platelets (TEPs) as the most common liquid biopsy markers has produced interesting and encouraging results with the thorough development of novel molecular technologies for CRC. A new potential for CRC medication resistance, early detection, disease monitoring, and therapeutic response is created by liquid biopsy [39,112,113]. ctDNA analysis is especially promising because it can reveal ongoing molecular changes, the emergence of acquired resistance, and the genetic heterogeneity of cancer cell populations. In addition, it is simpler to detect than CTCs. ctDNA can be employed for early disease detection and solid tumor surveillance [39]. The majority of current research focuses on the study of ctDNA variations using next-generation sequencing technology, but this detection method is challenging to adopt because of the process’ complexity, lengthy turnaround time for detection, and high cost. ctDNA methylation was also suggested to be used as a blood biomarker for tracking tumor recurrence at the same time. In a recent study, six ctDNA methylation markers were utilized to determine the ctDNA status in CRC. These markers have significant potential for risk classification, guiding adjuvant chemotherapy, and recurrence monitoring [114].

To assess the degree of EMT and its advancement in CRC can be difficult, which is a reality that needs to be recognized. This is because the expression of EMT markers frequently depends on the related cell type and the beginning signaling pathways.

## 7. The Role of EMT in Tumor Stroma and CRC Stem Cells

Cancer cells rarely initiate EMT as a cell-autonomous process. The expression of EMT-TFs, which in turn coordinate the expression of different EMT program components, is induced on cancer cells by signals from tumor-associated reactive stroma [44]. The microenvironment of a tumor is composed of stromal and immune cells that secrete a broad array of cytokines, chemokines, and growth factors. These secreted factors induce EMT in nearby cancer cells in a bystander manner by directly activating various EMT-TFs or the expression of effector molecules that suppress the epithelial state and promote the mesenchymal state (Figure 3). Three main types of stromal cells are identified: angiogenic vascular cells, infiltrating immune cells, and cancer cell-associated fibroblasts (CAFs). These cells are involved in the remodeling of the extracellular matrix (ECM) and in accelerating the growth and spread of tumors. A substantial amount of data indicate that the nontumor cell types that are recruited and reside in the tumor stroma are essential for controlling the behavior of cancer cells [36,44,50].

In addition to the microenvironment, factors that have the potential to influence the prognosis of patients with early-stage CRC include epigenetics, colonic crypt cell type, tumor mutational burden, and neoantigens. Effective clinical translation requires robust, clinically pertinent biomarkers and assays, which may need to be developed in the future based on current research.

It has been demonstrated that TGF-β signaling is crucial for EMT, and that TGF-β controls the immune system resistance, tumor stroma, microenvironment, and metastasis in CRC. Clinical and experimental trials have demonstrated the efficacy of inhibiting the TGF-β signaling system in the treatment of CRC [78,116,117,118,119]. Since many cancers block the epithelial pathways that encourage tumor growth, TGF-β signaling can have prometastatic effects on the tumor microenvironment even when it does not interfere with the signaling in epithelial cancer cells. Because TGF-β signaling causes a strong cytostatic reaction in epithelial cells, it is thought to be a tumor suppressor pathway in the development of colon cancer. Approximately forty percent of CRC has acquired mutations in TGF-β pathway components that result in a loss of function around the adenoma-to-cancer transition [107].

Important stromal cells known as CAFs are crucial to the development of tumors. In CRC cells, CAFs stimulate stemness, drug and treatment resistance, metastasis, and EMT. Through the direct transfer of exosomes to CRC cells, CAFs cause a notable upsurge in the levels of miR-92a-3p in CRC cells. By directly blocking the activation of the Wnt/β-catenin pathway by FBXW7 and MOAP1, as well as by decreasing mitochondrial apoptosis, increased expression of miR-92a-3p leads to cell stemness, EMT, metastasis, and 5-FU/L-OHP resistance in CRC [44,54,96,120,121]. In CRC, the release of exosomes from CAFs promoted cell stemness and EMT, which in turn boosted resistance to treatment and encouraged cell invasion. In addition to elevating the expression of CD133 and CD44 and inducing the EMT phenotype in CRC cells, which led to metastasis and heightened resistance to chemotherapy, CAFs also raised the percentage of CSC cells positive for these markers [120].

It has been thoroughly researched how macrophages cause EMT by secreting certain cytokines and chemokines. Tumor-associated macrophages (TAMs) secrete TGF-β, which acts similarly to TGF-β secreted by CAFs. Macrophages are involved in the transformation of colonic epithelial cells, and the interaction with macrophages’ glucose metabolism in the TME induces the “reverse Warburg effect” phenotype of enterocytes and accelerates tumorigenesis [44,122].

Mesenchymal cells, characterized by a spindle-shaped morphology like that of fibroblasts, can be induced in colon cancer cell lines by IL-1β. The morphological change and the IL-1β-treated cells’ decreased expression of E-cadherin occurred at the same time. Furthermore, IL-1β-induced EMT cells had a greater migratory capacity as compared to parental cells that had an epithelial phenotype, which could be involved in the development, spread, and recurrence of colon cancer. Moreover, IL-1β enhanced colon cancer cells’ ability to self-renew, which is a crucial characteristic of CSCs [123].

There is accumulating evidence to suggest that there is a crosstalk between tumors and their milieu, such as stromal cells, tumor microvasculature, extracellular matrix (ECM), and hypoxic microenvironment. This crosstalk can generate EMT and CSC features in cells, which can then contribute to the development of malignancies.

Since their initial discovery, research on CRC stem cells has uncovered previously unknown characteristics, such as a high degree of heterogeneity and plasticity. A major obstacle to the eradication of cancer is the ability of CRC stem cells to proliferate, withstand chemotherapy, and constantly adapt to a changing microenvironment by taking advantage of a confluence of genetic, epigenetic, and environmental variables [124].

The development and maintenance of CRC liver metastases are thought to be largely dependent on CSCs, which are thought to be the driving force behind tumor progression and metastasis [109,125]. CSCs are considered to drive cancer progression and to be the cellular seeds of tumor metastasis and disease recurrence, two of the most difficult challenges in clinical oncology, because they instigate tumor growth and have the apparent ability to resist multiple therapies [48]. CSCs have been identified in various malignancies using a combination of cell surface antigens [86].

Eliminating cancer stem cells, which are at the apex of a hierarchy prevalent in numerous cancer types, is regarded as a crucial aspect of effective antitumor treatment [126]. Precancerous stem cells (pCSCs) undergo complicated genetic and molecular alterations when exposed to diverse microenvironmental variables, leading to the slow transformation of precancerous stem cells into colon cancer stem cells, which is assumed to be the origin of most colorectal malignancies [127].

The presence of CSCs in CRC was first demonstrated in human-on-mouse tumor xenograft experiments. This cell fraction possessed enhanced clonogenic and tumorigenic abilities, which were later confirmed by mouse lineage tracing studies. Human CRC stem cells substantially contribute to clinical tumor progression, chemosensitivity, and treatment failure [128].

Cancer stem cells are thought to be a diverse group of cells that reside within tumors and are responsible for metastasis, tumor growth, and resistance to treatment. The CSC model of cancer regards tumor development as a malleable process in which normal cancer cells can dedifferentiate into CSCs, for instance, in response to cellular stress or therapy. From there, CSCs can differentiate into any of the other cell types that can be found in a particular tumor, including fibroblasts associated with cancer, vascular endothelial cells, and tumor-associated stem cells. EMT and MET are closely related to this differentiation and dedifferentiation [89,129]. Cancer cells that acquire EMT features also gain CSC-like properties, which often undergo EMT to create metastasis. The EMT process is linked to the acquisition of stem cell properties by both normal and cancer cells [44,47,54,109,123,130].

Metastatic CSC development can be directly facilitated by the EMT program through increased tumorigenesis. Moreover, cancer cells can transition between epithelial and mesenchymal cells due to inherent EMP [47,57]. After epithelial cells have undergone at least a portion of EMT, they are prepared to reach the epithelial stem cell state [47]. Both the EMT and MET induction programs must be balanced in order to establish and sustain a mixed state, and this balance is upheld by numerous feedback loops at the transcriptional, transduction, and epigenetic levels. Overall, the combination of ongoing EMP dynamics and a persistent E/M mixed phenotype led to a significant increase in the number of stem cells present in both normal and cancer cell populations [89]. A subpopulation of CSCs (CD133-high/CD26-high) in colon cancer clinical samples showed signs of activation of the EMT program, including increased expression of N-cadherin and vimentin, and decreased expression of E-cadherin [50].

Since the EMT process is what leads to the state of CCSCs, one more therapeutic strategy could be to use EMT inhibitors to block TGF-β and Wnt signaling, which is the most distinctive pathway known to positively effect this process [48,50,78]. The SMAD route and non-SMAD-mediated signaling pathways are two ways in which TGF-β is a powerful inducer of EMT, influencing gene expression alterations and activating EMT transcription factors [43,131]. The development of metastatic potential in CRC may involve dedifferentiation brought on by carcinogens and the acquisition of stemness characteristics through the loss of APC or AXIN tumor suppressors, which would activate the WNT/β-catenin signaling pathway. Macrophage inhibitory cytokines (MICs) are vulnerable to the cytostatic effects of TGF-β when the TGF-β-responsive SMAD4 tumor suppressor transcription factor is deactivated. This provides an opportunity for CAFs to release and activate excess TGF-β. TGF-β is the primary cause of CRC metastasis. It creates a positive feedback loop by consolidating CAF activation and encourages MICs to take on mesenchymal and stem properties [45].

## 8. Targeting EMT in the Treatment of CRC

However, an E/M hybrid state is sufficient for the EMT program to be activated, rather than a full mesenchymal phenotype. A higher probability of developing stem cell characteristics is seen in mixed EMT stages. Together with the stem cell state, or stemness window, the placement of cells along the EMT axis corresponds. There is growing evidence that suggests EMP may be a driver of cancer stemness since it regulates phenotypic plasticity in both cancer and normal cells. The identification of EMT and cancer stemness as primary factors contributing to treatment resistance renders this axis a desirable target for therapeutic intervention [89].

Wnt/β-catenin signaling is often aberrantly activated in CRC patients, and this is thought to be a key factor in the pathophysiology of CRC. It is possible to prevent tumor metastasis by blocking the Wnt/β-catenin pathway and survivin expression with the new Wnt/β-catenin inhibitor IWR-1. IWR-1 may therefore be taken into consideration as a therapeutic agent for the treatment of CRC in future clinical applications [59,100].

Multiple varieties of cancer have been found to contain CSCs that exhibit increased resistance to various existing treatments, such as radiation and chemotherapy. This resistance causes an increase in mesenchymal, stem-like cancer cells after the initial treatment, which frequently results in clinical relapse [47]. These factors help to explain why EMT and CSC programs are drawing more and more interest from individuals looking for fresh ideas in clinical treatment.

The prognosis of patients with CRC varies, with tumors on the right side having the worst prognosis according to the site of the primary tumor, so screening and early diagnosis strategies for these patients are particularly important [132]. It is difficult to develop clinical routes that target EMT modulators, effectors, or inducers because of the variability and plasticity of the several pathways involved. The utilization of EMT components as therapeutic targets is further complicated by spillover effects between pathways [40]. It is therefore doubtful that targeting a single EMT receptor will be successful due to the redundancy of many pathways; nevertheless, targeting upstream transcription factors may have a more significant impact.

From a therapeutic perspective, it would be beneficial to reverse the process of EMT, i.e., induce MET, given the critical role of the EMT program in the multiple malignant features of cancer. Cancer cells in a CSC state with an active EMT program will be compelled to undergo differentiation into non-CSCs and revert to epithelial features as a result, losing their increased proliferative potential and resistance to diverse treatments [50]. Inactivation of EMT and induction of MET are necessary for efficient metastatic colonization and growth at a distance. In this context, therapeutic promotion of MET may accelerate metastasis of disseminating cells. Therefore, the precise treatment period needs to be carefully defined. Loss of E-cadherin is a key feature of EMT, and restoring its expression may be considered a promising approach to suppress metastasis [40].

Approximately 30% of nonresponsive cases to anti-EGFR therapy result from unknown, seemingly nongenetic mechanisms of drug resistance, despite the fact that inherent genetic mechanisms, such as KRAS/NRAS/BRAF mutations and ERBB2 and MET amplifications, have been identified as contributing to resistance to anti-EGFR therapy. Though its role in cetuximab resistance in CRC is unclear, EMT is a crucial function that may be overridden by cancer cells and is linked to tumor aggressiveness and treatment resistance [59].

It has been shown that TGF-β can promote EMT through both SMAD-dependent and -independent mechanisms. Therefore, therapeutic strategies that target these signaling pathways through TGF-β may be effective in stopping cancer cells from growing invasively and from spreading [84]. Targeting CMS4-like TGF-β-activated CRC, clinical trials have started on the first treatment combination, opening the door to overcome the tumor microenvironmental interaction that maintains the mesenchymal phenotype [66].

By stimulating CSCs, IL-1β may encourage self-renewal and EMT, and ZEB1 is essential for these two processes. Therefore, ZEB1 and IL-1β could be novel therapeutic targets for stem cells that cause colon cancer [123]. The impact of miRNAs on clinical practice is anticipated within the next decade, as they represent the most promising and fastest-growing group of prospective CRC biomarkers [21].

By suppressing the Wnt signaling pathway in CRC, curcumin may be able to stop EMT. Curcumin has the ability to increase the expression of naked cuticle isotope 2 (NKD2), an inhibitor of the Wnt axis that is secreted. In addition, curcumin inhibits tumor invasion and motility and downregulates the expression of chemokine receptor 4 (CXCR4), indicating a potential mechanism to stop the evolution of CRC [59].

By preventing NF-κB from activating its promoter in CRC, LCN2 prevents the NF-κB /SNAIL signaling pathway from proliferating, spreading, and atrophy. The LCN2/ NF-κB /SNAIL pathway has thus been proposed as a novel prognostic and therapeutic approach for CRC [133].

Multiple lines of evidence indicate that EMT is an epigenetic process independent of alterations in the DNA sequence of normal and tumor cells. The expression of this program and its comprehensive impact on tumor biology cannot be determined by sequencing the genomes of cancer cells, as is the case with many other biological programs [44]. EMT appears to be a crucial consideration for improving and refining the current clinical classification of CRC and better stratifying patients for prognosis and therapy. However, the lack of reliable and robust biomarkers continues to restrict its application.

## 9. Conclusions

When considered as a whole, EMT plays a crucial part in the progression of CRC and is a prospective target for preventing colon cancer from acquiring invasiveness or avoiding recurrence and metastasis after its excision. Both of these outcomes are associated with the progression of the disease. Therefore, modification of the several pathways that target EMT could be a viable therapeutic strategy for CRC.

In clinical settings, the identification and development of EMT-associated biomarkers has been hampered by the difficulties of correctly recognizing the EMT process at the morphological level. However, the development of liquid biopsy, which includes CTCs and ctDNA, has opened up a new door in the early detection of CRC, as well as in the monitoring of the disease, the treatment response, and the development of drug resistance.

CMS4 has the worst prognosis out of the four subtypes of CRC, and therapy that targets CMS4-like TGF-β-activated CRC paves the route for overcoming the tumor microenvironmental interaction that promotes the mesenchymal phenotype.

In the end, it will be important to identify a general consensus of robust, validated biomarkers in order to enhance the detection of, treatment for, and prognosis of CRC.

## Figures and Tables

**Figure 1 ijms-24-14815-f001:**
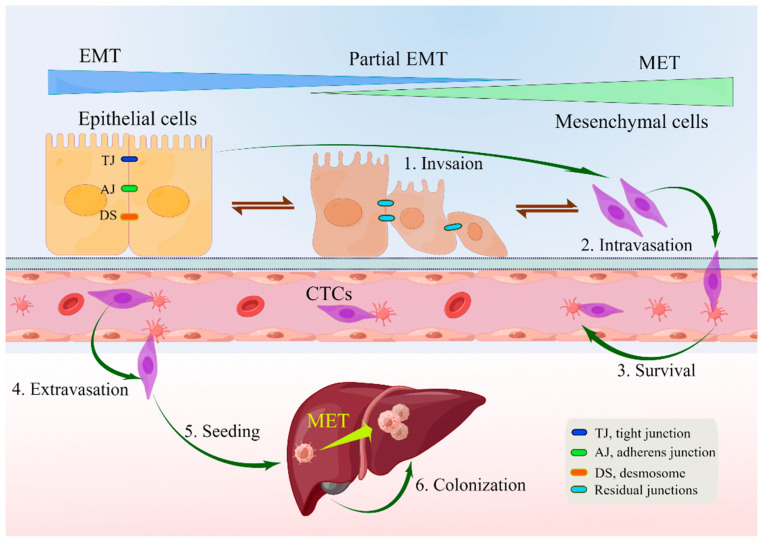
Depiction of EMP and invasion–metastasis cascades in CRC cells [22,40,45]. Along the entire EMT spectrum, CRC cells have different degrees of growth and metastasis. Cells can move along these two states with flexibility since EMT and MET are nonbinary reversible processes. Malignant cells were those that exhibited more epithelial characteristics and less mesenchymal transition. Primary CRC cells undergoing EMT undergo a series of intermolecular changes that lead to loss of intercellular adhesion, including dissolution of intercellular junctions, namely tight junctions (dark blue), adherens junctions (green), and desmosomes (red orange); through this process, cells acquire a mesenchymal phenotype that promotes local migration and invasion. The cells then enter the blood vessels (intravasation), survive in the blood vessels as circulating tumor cells, and leave the blood vessels (extravasation) to seed and colonize the parenchyma of the liver. After seeding locally in the liver, cancer cells undergoing EMT can redifferentiate to an epithelial phenotype through MET, a step that facilitates cell colonization in the liver and development of local metastases. Abbreviations: EMP, epithelial–mesenchymal plasticity; CTC, circulating tumor cells; EMT, epithelial–mesenchymal transition; MET, mesenchymal–epithelial transition.

**Figure 2 ijms-24-14815-f002:**
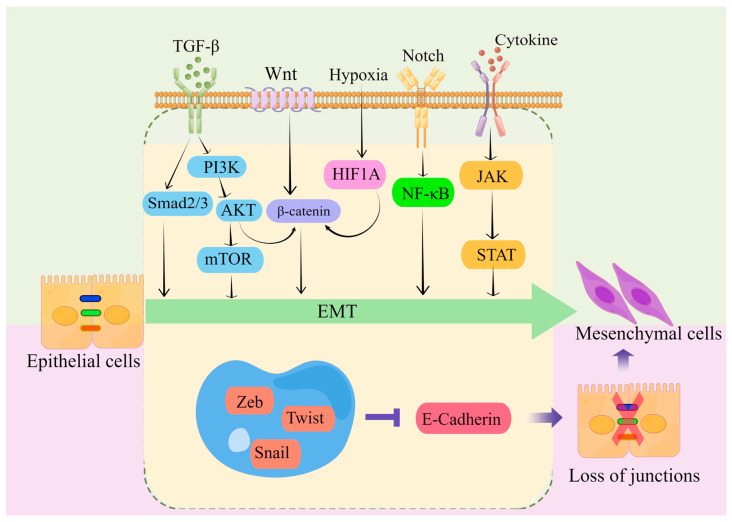
Key pathways regulating EMT-TFs during EMT development in CRC [44,55,83,84,85,86,87]. There are several EMT-TFs (such as ZEB, SNAIL, and TWIST) that cooperate with signaling pathways in cells, and under the stimulation of factors such as TGF-β, Wnt, Notch, hypoxia, etc., repress genes related to epithelial status (such as E-cadherin), which leads to the loss of intercellular connections and allows epithelial cells to differentiate into a mesenchymal state. These transcription factors are pleiotropic, inducing a transition to a mesenchymal or partially mesenchymal cellular state. Abbreviations: EMT, epithelial–mesenchymal transition; PI3K, phosphatidylinositol-3-kinase; AKT, protein kinase B; mTOR, mammalian target of rapamycin; HIF1A, hypoxia inducible factor 1 subunit alpha; NF-κB, nuclear factor kappa B; JAK, janus kinase; STAT, signal transducer and activator of transcription; ZEB, zinc finger E-box-binding homeobox.

**Figure 3 ijms-24-14815-f003:**
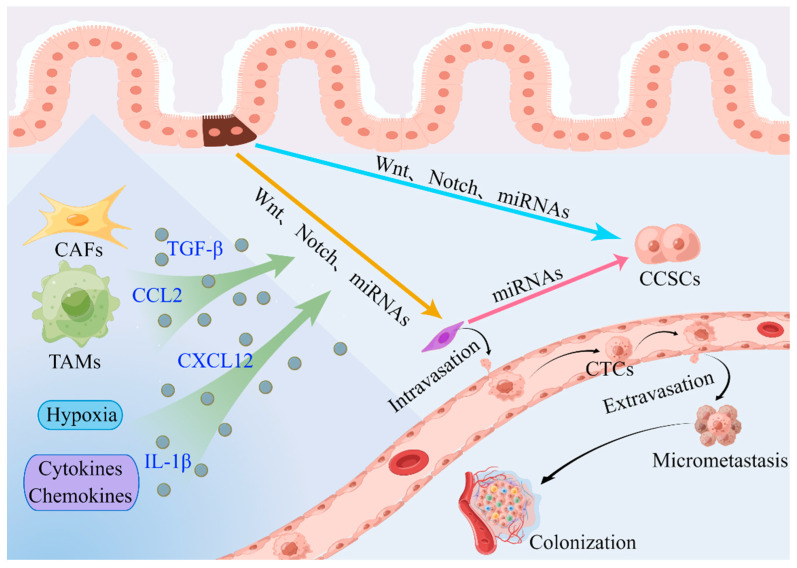
Contribution of the tumor microenvironment to the activation of the EMT program and the invasion–metastasis cascade in CRC [50,83,84,115]. CAFs, TAMs, hypoxia, cytokines, and chemokines make important contributions to the induction of EMT in nearby cancer cells and the invasion–metastasis cascade in CRC. EMT cells have cancer stem cell-like characteristics, and CSCs exhibit a mesenchymal phenotype. Aberrant miRNA expression correlates with CSC formation and acquisition of the EMT phenotype. Abbreviations: CAFs: cancer-associated fibroblasts; TAMs: tumor-associated macrophages; CTCs: circulating tumor cells; CCSCs: colorectal cancer stem cells.

**Table 1 ijms-24-14815-t001:** Comparison of differences between epithelial and mesenchymal cells.

Feature	Epithelial Cells	Mesenchymal Cells
Morphology	Polygonal, pebble-like shape	Elongated, spindle-shaped [47]
Polarity	Apical–basal	Front–back [50]
Invasion ability	No mobility	Enhanced movement and raiding capabilities [55]
Cytoskeleton	Express cytokeratins	Express vimentin
Intercellular junction	Tight junctions, adherens junctions, and desmosomes	Loosely attached to the extracellular matrix [55]
Interactions with extracellular matrix	Via integrin α6β4 at hemidesmosomes; linked to cytokeratins	Via β1-containing or β3-containing integrins at adhesion plaques; linked to actin stress fibers [44,50]
Markers	Junctional proteins, epithelial markers, matrix proteins	Matrix proteins, proteases, mesenchymal markers [56]

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
