# Peer review of "Role of Epithelial to Mesenchymal Transition in Colorectal Cancer"

_ijms, 2023, doi:10.3390/ijms241914815_

Round 1
Reviewer 1 Report
Jian Lu et al. demonstrated the role of epithelial-mesenchymal transition in colorectal cancer. The review is very impressive and well written. I deeply congrats to the authors on this nice review. However, there are minor corrections that should be addressed.
- The authors should provide the references according to the journal instructions.
- Abstract should be only one paragraph.
- All the abbreviations should be revised, and a list of abbreviations should be provided.
- Table 1 references should be embedded in the table.
- Page 6: TGF- is an extensively researched cytokine associated with EMT" should be corrected, and I also recommend the complete revision of the review.
Author Response
|
Response to Reviewer 1 Comments
|
||
|
1. Summary |
|
|
|
Dear reviewer, thank you very much for taking time to review this manuscript. I am grateful for your high evaluation of this review and your suggestions for modifications make this review more standardized. I have modified some places according to your suggestions one by one. |
||
|
2. Questions for General Evaluation |
Reviewer’s Evaluation |
Response and Revisions |
|
Does the introduction provide sufficient background and include all relevant references? |
Yes |
|
|
Are all the cited references relevant to the research? |
Yes |
|
|
Is the research design appropriate? |
Not applicable |
|
|
Are the methods adequately described? |
Yes |
|
|
Are the results clearly presented? |
Not applicable |
|
|
Are the conclusions supported by the results? |
Not applicable |
|
|
3. Point-by-point response to Comments and Suggestions for Authors |
||
|
Comments 1: Jian Lu et al. demonstrated the role of epithelial-mesenchymal transition in colorectal cancer. The review is very impressive and well written. I deeply congrats to the authors on this nice review. However, there are minor corrections that should be addressed. 1.The authors should provide the references according to the journal instructions. 2.Abstract should be only one paragraph. 3.All the abbreviations should be revised, and a list of abbreviations should be provided. 4.Table 1 references should be embedded in the table. 5.Page 6: TGF- is an extensively researched cytokine associated with EMT" should be corrected, and I also recommend the complete revision of the review.
|
||
|
Response 1: Dear reviewer, thank you very much for your suggestions. I have been using revision mode to make point-by-point revisions in the original text, and every one of your suggestions has been properly addressed. In addition, I have made moderate language polish to some of the original texts. I replaced three illustrations from the original review, and they were all replaced with non-watermarked, authorized images. |
||
|
4. Response to Comments on the Quality of English Language |
||
|
Point 1: |
||
|
Response 1: Thank you for your recognition of the language quality of this review. |
||
|
5. Additional clarifications |
||
Reviewer 2 Report
Authors of the manuscript »Role of Epithelial to Mesenchymal Transition in Colorectal Cancer« have performed a review of literature published in the period of 2008-2023 on the subject matter.
In the introduction section they have described the clinical burden of colorectal cancer in developed world an outlined the key aspects of its pathophysiology. In the following sections they have thoroughly elaborated the physiological role of EMT in tissue development and its crucial role in epithelial cancer. They have nicely explained the interplay of important adhesion molecules and involved signaling pathways in the EMT process. Throughout the manuscript they were able to show the immense complexity of the EMT in colorectal cancer and its important role in the metastasing process. They also nicely explained the concept of tumor heterogeneity also in terms of EMT completeness with the introduction of partial EMT and hybrid E/M state. The broader picture on tumor biology has been provided with explanation of the key cellular players of tumor microenvironment as well as tumor stem cells. Insights on potential clinical implications are also given to the reader. The authors conclude the rather long and comprehensive manuscript with reaffirmation of the importance of EMT in the progression of colorectal cancer and realization of unavailability of clinically useful biomarkers for its assessment. The manuscript is supported with extensive amount of relevant references.
Author Response
|
Response to Reviewer 2 Comments |
||
|
1. Summary |
|
|
|
Dear reviewer, thank you very much for taking time to review this manuscript. I am grateful for your high evaluation of this review. |
||
|
2. Questions for General Evaluation |
Reviewer’s Evaluation |
Response and Revisions |
|
Does the introduction provide sufficient background and include all relevant references? |
Yes |
|
|
Are all the cited references relevant to the research? |
Yes |
|
|
Is the research design appropriate? |
Not applicable |
|
|
Are the methods adequately described? |
Yes |
|
|
Are the results clearly presented? |
Not applicable |
|
|
Are the conclusions supported by the results? |
Not applicable |
|
|
3. Point-by-point response to Comments and Suggestions for Authors |
||
|
Comments 1: Authors of the manuscript »Role of Epithelial to Mesenchymal Transition in Colorectal Cancer« have performed a review of literature published in the period of 2008-2023 on the subject matter. In the introduction section they have described the clinical burden of colorectal cancer in developed world an outlined the key aspects of its pathophysiology. In the following sections they have thoroughly elaborated the physiological role of EMT in tissue development and its crucial role in epithelial cancer. They have nicely explained the interplay of important adhesion molecules and involved signaling pathways in the EMT process. Throughout the manuscript they were able to show the immense complexity of the EMT in colorectal cancer and its important role in the metastasing process. They also nicely explained the concept of tumor heterogeneity also in terms of EMT completeness with the introduction of partial EMT and hybrid E/M state. The broader picture on tumor biology has been provided with explanation of the key cellular players of tumor microenvironment as well as tumor stem cells. Insights on potential clinical implications are also given to the reader. The authors conclude the rather long and comprehensive manuscript with reaffirmation of the importance of EMT in the progression of colorectal cancer and realization of unavailability of clinically useful biomarkers for its assessment. The manuscript is supported with extensive amount of relevant references. |
||
|
Response 1: Dear reviewer, thank you very much for your suggestions. I have made moderate language polish to some of the original texts. I replaced three illustrations from the original review, and they were all replaced with non-watermarked, authorized images. |
||
|
4. Response to Comments on the Quality of English Language |
||
|
Point 1: |
||
|
Response 1: Thank you for your recognition of the language quality of this review. |
||
|
5. Additional clarifications |
||